# Impacts of Harvest Year and Cultivation Location on Off-Flavor Compounds and Functionality of Pea Protein Isolate

**DOI:** 10.3390/foods13213423

**Published:** 2024-10-27

**Authors:** Tamar Manouel, Busra Gultekin Subasi, Mehdi Abdollahi

**Affiliations:** Department of Life Sciences, Food and Nutrition Science, Chalmers University of Technology, SE-41296 Gothenburg, Sweden

**Keywords:** plant protein, pea protein extraction, off-flavors, lipoxygenase, beany flavors

## Abstract

The impact of four different harvest years and two cultivation locations (CLs) of pea seeds on their protein wet fractionation yield, volatile and non-volatile beany flavors, and functionality were investigated. Both harvest years and CLs significantly affected protein recovery, but protein purity was primarily influenced by CLs. Seed age emerged as a dominant factor causing the reduction in linolenic/linoleic acid content and lipoxygenase (LOX) activity which surpassed the effect of harvest years in the seeds but not in their proteins. CL significantly affected fatty acid composition in both seeds and proteins, whereas its effect on LOX activity was discernible only in the proteins. Volatile beany compounds in the proteins were affected by both harvest years and CLs, correlating with their polyunsaturated fatty acid (PUFA) content and LOX activity. Both factors minimally impacted the emulsification capacity of the proteins but imposed a significant effect on their rheological properties. Altogether, the results revealed that seed crop years and especially locations affect pea protein quality, calling for proper adaptation strategies.

## 1. Introduction

Concerns about the environmental and ethical issues of animal farming, the rapidly growing global population, and the rising popularity of vegetarian and vegan diets have all contributed to the increased demand for plant-based proteins [1,2]. Among plant protein sources, field peas are highly recognized for their high-quality protein and nutrient content. They are widely available, particularly in cold climates, and are affordable with low allergenic potential. Additionally, field peas offer proteins with favorable nutritional and functional properties [3]. However, field peas as a protein source present challenges due to rapid lipid oxidation, which can lead to beany flavors or off-flavors. Additionally, the presence of antinutrients further complicates this issue [4]. These off-flavors limit the application of pea proteins in meat and dairy alternatives, where a neutral taste is essential for consumer acceptance [5].

The off-flavors of pea protein, which are also known as “beany, grassy and astringency”, are largely associated with lipid degradation as a consequence of chemical and enzymatic reactions that occur during harvesting, storage, and processing [6]. The typical “beany” off-flavor is associated with “green” and “earthy” aroma perceptions, resulting from a combination of various volatile organic compounds (VOCs). These small organic molecules belong to several chemical families, including aldehydes, ketones, alcohols, and furans. Notable components contributing to this off-flavor include hexanal, heptanal, (E, E)-3,5-octadien-2-one, pentanol, (E)-2-octenal, and pentylfuran, among others [7]. Furthermore, a major cause of off-flavor formation is the oxidation of unsaturated fatty acids, such as linoleic acid—the predominant fatty acid in dry whole peas—by the enzyme lipoxygenase (LOX). Non-volatile compounds (non-VOCs) also contribute to the off-taste of pea protein due to bitterness-inducing compounds like saponins and polyphenols, which are linked to the bitter taste and astringency in peas. [8].

The generation of ‘beany’ off-flavor initiates in the very early stages right after harvesting and can develop during storage due to the rapidly propagating lipid oxidation in the fresh raw material. Thereafter, lipids are quickly converted to fatty acid hydroperoxides (HPODs), forming volatile and non-volatile compounds [5]. Previous studies have indicated that various factors, such as pea cultivar, morphology, mechanical harvesting, and storage conditions, can influence the rate of lipid oxidation and the formation of volatile compounds in peas. These factors affect the composition of peas, the content and activity of LOX, its access to fatty acids, and the rate of chemical reactions [9]. It has been also shown by Nikolopoulou et al. (2007) [10] that harvest location and year have a big impact on not only the crude composition (protein, fat, and ash) of field pea seeds but also their sucrose, starch, and non-starch polysaccharide contents, as well as the total tannin and phytic acid contents. Later, Nosworthy et al. (2021) [11] showed that the cultivation year, due to variations in environmental conditions, significantly influences pea seed crude composition and amino acid profile as well as in vitro protein digestibility. Despite the studies on pea seeds, the impacts of harvest location and cultivation year on the content and formation of volatile and non-volatile off-flavor compounds in pea proteins and their functionality remain unknown, to the best of our knowledge.

This study aimed to investigate the influence of different harvest years and locations on beany flavor formation in pea flour and crude pea protein isolate. This was achieved by investigating the key volatile compounds associated with beany flavors and non-volatile off-flavor compounds including saponins, total phenolic compounds, and tannins together with the lipoxygenase enzyme activities. Yellow pea samples sourced from four different harvesting years and two locations were processed in the same year into crude protein isolates using alkaline solubilization and isoelectric precipitation. In addition, the effect of harvest years and locations on the protein yield, composition, molecular structure, and functional properties of the crude isolates were evaluated.

## 2. Materials and Methods

### 2.1. Materials

Field pea (*Pisum sativum* L.) samples from the Ingrid variety cultivated at the same location called “Svalöv” but in different harvest seasons of 2018, 2019, 2020, and 2022 were provided by Lantmännen AK. The samples, planted in specific lots and harvested each year, were stored in the dark at 4 °C at Lanmännen before their transfer to the facilities of Chalmers University of Technology 2022 to be analyzed at the same time. Additionally, field pea samples from two distinct locations, “Svalöv and Kölbäck”, harvested in 2022 were also provided by Lantmännen AK and had been harvested and treated the same as the other samples. They are referred to as Locations 1 and 2, respectively. All the samples were stored in the dark at 4 °C until the analysis.

### 2.2. Pea Sample Pre-Treatment and Protein Extraction Process

Whole dry pea samples were subjected to dehulling using a Satake TM05 abrasive mill (Satake, Hiroshima, Japan), employing a technique outlined in [12], followed by the separation of hulls and seeds. The obtained split peas were then ground into flour using a Retsch ZM 200 ultra-centrifugal mill (Retsch, Haan, Germany), equipped with a 500 µm screen at 12,000 rpm, following the approach introduced by Gu et al. (2021) [13]. The ground pea flour was stored in the dark at 4 °C in plastic bags until a further protein extraction process.

Pea proteins were then isolated from each respective flour using alkaline solubilization and isoelectric precipitation in duplicate. For each treatment, approx. 60 g of flour was dispersed in distilled water at a ratio of 1:15 (*w*/*v*), and its pH was adjusted to 8.5 using 2 M NaOH. The solution was stirred for 60 min at room temperature with the pH being checked at a 15 min interval and readjusted to maintain it as constant, if necessary, according to the method presented in [14,15]. This process was followed by subjecting the sample to centrifugation at 4000× *g* for 20 min at 20 °C and collecting the supernatant containing the solubilized proteins. Thereafter, the precipitation step was employed with an acid treatment by adding 2 M HCl to reach pH 4.5 and holding for 10 min. This step was also followed by a second centrifugation at 4000× *g* for 20 min at 20 °C to collect the isolated proteins. The pellets were re-suspended in distilled water (1:1, *w*/*v*) and their pH was readjusted to 7.0 with 2 M NaOH. The protein slurry at pH 7.0 was then frozen at −80 °C, followed by freeze-drying at 0.1 to 0.5 mbar for 24 h. The crude pea protein isolates were then stored in zipped plastic bags in the dark at 4 °C for further analysis.

The mass yield and protein recovery of the protein isolation process were calculated using the following equations:$$\mathrm{Mass}\;\mathrm{yield}\;(\%)=\frac{Amount\;of\;dry\;protein\;isolate}{Amount\;of\;dry\;starting\;material}\;\times \;100$$
$$\mathrm{Protein}\;\mathrm{recovery}\;(\%)=\frac{Amount\;of\;final\;product\times protein\;content}{Amount\;of\;starting\;material\times protein\;content}\;\times \;100$$

### 2.3. Proximate Compositions and Color Measurement

The protein content of both pea flours and protein isolates was determined by the Dumas method with a nitrogen analyzer (Elementar Analysensysteme GmbH, Langenselbold, Germany), and a conversion factor of 6.25 was used to calculate the total protein content. The total fat content was determined based on a modified method from Undeland et al. (2002) [16] and Yoshida et al. (2007) [17]. Approximately, 0.5 g of each sample (pea flours and protein isolates) was mixed with 20 mL of an ice-cold chloroform/methanol (2:1) solution with 0.1 butylated hydroxytoluene (BHT). Thereafter, 50 μL of an internal standard (C17) was added, followed by shaking for 30 min. Finally, 8 mL 0.5% NaCl was added to each sample for clarification and vortexed for 30 S. The final solution was centrifuged for 6 min at 3000× *g*, at 4 °C, and the bottom part was transferred for evaporation under mild conditions, at 30 °C for 90 min. The samples were then weighted, and the lipid content was calculated gravimetrically.

Moisture content was determined by a moisture analyzer (Mettler Toledo, HE73, Greifensee, Switzerland) at 110 °C. For ash content determination, pea flour and protein isolate samples were oven-dried at 105 °C for 16 h, followed by cooling down to room temperature in a desiccator before transferring them into an ashing furnace at 550 °C for 16 h, programmed according to (AOAC, 2005) [18].

The surface color of pea flours and protein isolates was identified using a handheld colorimeter (Croma Meter CR-400, Konica Minolta, Tokyo, Japan). The color was measured according to the Hunter color measurement system. The result was given in values for L (lightness), a (green–red), and b (blue–yellow), and from this, yellowness and whiteness values were calculated according to the following formulas:
YI_FC_ = 142.86 × b/L
$$\mathrm{Whiteness}=100-\sqrt{{\left(100-L\right)}^{2}+a^{2}+b^{2}}$$

### 2.4. Lipoxygenase Activity Measurement

The measurement of *lipoxygenase* (LOX) values in the samples was conducted through a method explained in [19,20]. First, 10 mL of phosphate buffer (10 mM, pH 6.5) was added to 0.1 g of each of the pea flours and protein isolates and then stirred magnetically at room temperature for 5 h. The mixture was then centrifuged at 9100× *g* for 10 min, and the supernatant was used as the crude enzyme extract for further analysis. The substrate solution was prepared by a mixture of linoleic acid (140 μL) and Tween (140 μL) and emulsified with 8 mL phosphate buffer (pH 6.5). Then, 1.1 mL of 0.5 M NaOH was added to clarify the solution, and the volume was brought to 50 mL with phosphate buffer. The stock substrate solution was flushed with nitrogen to prevent any oxidation and was diluted (1:40, *v*/*v*) with 0.2 M sodium borate buffer (pH 9.0) before use. Thereafter, 50 μL of the crude enzyme extract was added to 1.25 mL substrate with rapid mixing for 5 sec, and the change in absorbance increased due to the presence of a conjugated hydroperoxide moiety of the mixture at 234 nm was recorded for 3 min using a UV–visible spectrophotometer (Cary 60UV-vis, Agilent Technologies, Santa Clara, CA, USA). The unit of LOX activity was U/g, where U is defined as the numeric increase in absorbance per minute with the following equation:$$LOX\;activity\left(\frac{unit}{g}protein.min\right)=\frac{∆ABS\times 1000\times 1000}{Protetin\;content\;(mg)\times 3}$$

The numbers 1000 and 1000 are the LOX unit conversion (0.001 ABS indicates 1 unit) and the mg to g conversion, respectively. And the number 3 is the total assay time in min.

### 2.5. Fatty Acid Methyl Ester Analysis

The fat extracts collected after the determination of the total fat content were methylated by adding 1 mL toluene and 1 mL methanol/acetyl chloride (10%, *v*/*v*) solution and incubated for 120 min at 60 °C as described by Subasi et al. (2023) [5]. After that, 1 mL of Milli-Q water and 1.5 mL petroleum ether were added and vortexed for 20 S. The solution was then centrifuged at 2500× *g* for 5 min, and the upper part was transferred and evaporated with N_2_ at 40 °C until complete dryness. The evaporated samples were re-dissolved in 500 μL isooctane to be used for GC-MS injection. The fatty acids were identified and quantified using GC-MS analysis carried out on an Agilent 7890A GC system, equipped with a VF-wax column and coupled with an Agilent 5975C triple-axis MS detector (Agilent Technologies, Santa Clara, CA, USA). Fatty acid methyl esters (FAMEs) were identified by comparing them to the GLC 463 standard (Nu-Check prep, Inc., Elysian, MN, USA). The fatty acids as linoleic, linolenic, oleic, and palmitic acids, in addition to saturated and unsaturated fatty acid variations, in both pea flours and protein isolates, were defined by the described extraction and methylation methods based on [21].

### 2.6. Analysis of Volatile Off-Flavor Compounds

Headspace Solid-Phase Microextraction coupled to Gas Chromatography–Mass Spectrometry (HS-SPME-GC–MS) was used for the identification of volatile off-flavor contributors, employing a method established by Sajib et al. (2023) [22] with some modifications. Around 1 g of the pea flours and 0.05 g of protein isolates were dissolved in 8 mL of Milli-Q water within 20 mL SPME vials. The volatiles from the vial headspace were collected by an SPME fiber (75 μm Carboxen/polydimethylsiloxane (CAR/PDMS), Supelco, Bellefonte, PA, USA) during a 40 min extraction at 60 °C under continuous stirring (500 rpm). Subsequently, the adsorbed volatile compounds were thermally desorbed and injected into the GC-MS instrument for 5 min using the splitless injection mode. A Shimadzu TQ8030 GC–MS setup equipped with a ZB-1701 capillary column (30 m × 0.32 mm, 1 μm, Phenomenex, Shinagawa, Japan) was used for analysis, with data acquisition scans ranging from 30 to 500 amu. Helium was employed as the carrier gas at a flow rate of 1.5 mL/min. The GC inlet temperature was held at 300 °C, and the GC separation occurred within an oven with temperatures ranging from 35 to 260 °C. The MS transfer line temperature was set to 265 °C, while the ion source temperature was maintained at 200 °C. During this analysis, selected compounds of volatile off-flavors in peas were indicated based on the most commonly reported volatile beany flavor markers ( Sajib et al., 2023 [22]; Xu et al., 2019 [19]) including hexanal, 1-hexanol, 1-octen-3-ol, benzaldehyde, 2-pentylfuran, 1-pentanol, 1-nonanol, 2-methoxy-3-isopropyl pyrazine, (E,E)-2,4-nonadienal, (E,E)-2,4-decadienal, and 3-methyl-1-butanol. Quantification of volatile compounds was accomplished by calculating the relative peak areas of the targeted compounds against an external standard, using the peak area of the chosen internal standard as a reference.

### 2.7. Measurement of Non-Volatile Off-Flavor-Associated Compounds

To measure total saponins, pea flours and protein isolates (0.5 g) were mixed with 5 mL of 60% ethanol and put in a microwave digestor (Milestone microwave laboratory system; EthosPlus, Sorisole, Italy) as described by Subasi et al. (2023) [5]. The mixture was then centrifuged at 4000× *g* for 5 min. The supernatant was filtered with a 0.45 μm syringe filter and evaporated at 60 °C for approximately 2 h until complete drying. The dried extracts were dissolved in pure methanol at 2 mg/mL. Aliquots of 125 μL from the extract, 125 μL of freshly prepared vanillin in ethanol (8%, *w*/*v*), and 1.25 mL of sulfuric acid in water (72%, *v*/*v*) were transferred to the vials. After that, the samples were vortexed and heated at 60 °C for 10 min, and the tubes were transferred to a beaker full of ice to immediately cool down to room temperature. Total saponin content (TSC) was determined colorimetrically by measuring the absorbance of the mixture at 520 nm using a UV-VIS spectrophotometer (Cary 60UV-vis, Agilent Technologies, Santa Clara, CA, USA). Oleanolic acid was used as a saponin standard (50–800 μg/mL). The results were expressed as g of total saponins per 100 g extract [22,23].

The total phenolic content (TPC) was determined according to the method described in [13]. For this purpose, 0.5 g of the pea flours and protein isolates were mixed with 3 mL of 75% methanol and sonicated in an ultrasonic bath (USC900TH type, VWR international Ultrasonic Cleaners, Radnor, PA, USA) at room temperature for 15 min. The solution was then centrifuged at 2500× *g* for 100 min and 4 °C and the supernatant was collected. The extraction was repeated with the addition of 2 mL of 75% methanol to the pellet, followed by the centrifugation and collection of the supernatant. The total phenolic content of methanolic extracts was determined according to the method of Singelton and Rossi (1965) [24], by mixing 100 μL of the sample with 750 μL of Folin Ciocalteu reagent. After that, 750 μL of 6% sodium carbonate solution was added to the mixture and incubated for 60 min in the dark. The absorbance was measured at 765 nm with a UV-VIS spectrophotometer, and the results were calculated with the calibration curve prepared with gallic acid and were expressed as mg gallic acid equivalent (GAE) in 100 g of protein isolate.

The total tannin content (TTC) quantification was performed using a method developed in [25]. The determination was performed in two steps, first determining the total phenolic content and then removing the tannin from the extracts. Tannin-containing extract/sample was mixed with distilled water (1:5 *v*/*v*). Following that, 100 mg of polyvinyl polypyrolidone (PVPP) was added to 1 mL of distilled water and mixed with 1 mL of tannin-containing extract. The mixture was vortexed and incubated at 4 °C for 15 min and then centrifuged at 3000× *g* for 10 min, and the supernatant that contained only simple phenolics other than tannins was collected. The absorbances of the tannin-containing extracts and the non-tannin-containing extracts were measured at 765 nm with a UV-VIS spectrophotometer (Cary 60UV-vis, Agilent Technologies, Santa Clara, CA, USA). The total tannin content was then calculated by subtracting the non-tannin from the total phenolic content, and their difference was recorded as mg TAE in 100 g of the sample on a dry basis.

### 2.8. Analysis of Functional Properties

#### 2.8.1. Emulsification Capacity

The emulsifying activity index (EAI) and emulsion stability (ES) were determined based on the [14] method. A solution containing 150 mg of each pea protein isolate and 15 mL deionized water was mixed dropwise with 5 g of sunflower while the mixture was homogenized at a speed of 8000 rpm for 5 min, using a Polytron homogenizer (T18 Ultraturrax, IKA, Staufen, Germany) in the presence of an ice chamber to regulate any significant temperature rise. After homogenization, 50 μL of the emulsion was transferred from the bottom of the container at 0 and 10 min and mixed with 5 mL of 0.1% SDS solution. The blend was vortexed for 5 s, and then the absorbance of the diluted solution transferred to was measured at 500 nm using a UV-VIS spectrophotometer (Cary 60UV-vis, Agilent Technologies, Santa Clara, CA, USA) with 1 cm path length plastic cuvettes. The emulsifying activity index (EAI) values were calculated using the following equation:$$EAI\left(\frac{m^{2}}{g}\right)=\frac{2\times 2.303\times A_{0}\times DF}{C\times \varphi \times \theta \times 10000}$$

The emulsion stability (ES) values were calculated using the following equation:$$ESI\;\left(min\right)=\frac{A_{10\;}\times ∆t}{∆A}$$
where A_0_ is the measured absorbance at 500 nm, A_10_ is the absorbance at 10 min after homogenization, DF is the dilution factor, θ is the cuvette path length, C is the initial concentration of protein (g/mL), φ the volume fraction of oil in the emulsion, ∆t is the elapsed time that is 10 min, and the ∆A is the difference in absorbance values between initial and final readings (A_0_–A_10_).

#### 2.8.2. Rheological Analysis (i.e., In Situ Gelation)

For rheological (i.e., in situ gelation) analysis, 6 g of each protein isolate was mixed with 24 mL of deionized water. After that, the rheological analysis was performed as described in [26]. Approximately 1–2 g of each protein isolate from the chopper was introduced onto the dynamic rheometer (Paar Physica Rheometer MCR 300, Anton Paar GmbH, Graz, Austria) for in situ gelation analysis. This analysis employed a parallel-plate geometry (25 mm plate diameter and 1 mm plate gap) mounted on the dynamic rheometer, which operated in oscillating mode. To prevent evaporation during the in situ gelation process, the edges of the sample were shielded with inorganic oil. The in situ gelation procedure involved three steps: firstly, the temperature was gradually increased from 20 °C to 90 °C at a constant heating rate of 5 °C/min; next, the temperature was maintained at 90 °C for 30 min; finally, the temperature was reduced from 90 °C back to 20 °C at a rate of 5 °C/min. During the gelation test, measurements were conducted within the linear viscoelasticity region of the samples, specifically at a strain of 1% and a frequency of 0.1 Hz.

### 2.9. Sodium Dodecyl Sulfate–Polyacrylamide Gel Electrophoresis (SDS-PAGE)

The protein profiles of the polypeptides within the flour and protein isolate samples from various pea varieties were ascertained using SDS-PAGE, following the methodology outlined by Laemmli (1970) [27]. Electrophoresis was executed utilizing Mini-protean TGX 4–20% pre-cast gels (Bio-Rad Laboratories, Hercules, CA, USA). Each sample was diluted with 0.1 M NaOH and mixed in a 1:1 ratio (*v*/*v*) with a Laemmli sample buffer supplemented with 2-mercaptoethanol, resulting in a final protein concentration of 2 µg/µL. Subsequently, 10 µL of each protein solution was loaded into individual wells per sample. A polypeptide standard (10–250 kDa) from Bio-Rad Laboratories (USA) was employed for band identification. For protein visualization, the gel bands were stained using a mixture of Coomassie Brilliant blue (0.02% *w*/*v*) in 50% methanol and 7.5% acetic acid (*v*/*v*) for 45 min, followed by destaining in a mixture of 50% methanol and 7.5% acetic acid (*v*/*v*) for 90 min. Gel imaging was carried out using Bio-Rad’s Gel Doc 2000 after overnight refrigeration.

### 2.10. Statistical Analysis

All data were expressed as mean ± standard deviation (SD) (n = 2 or 3). One-way ANOVA and Duncan post hoc tests were used to determine significant differences. The level of significance was set to *p* < 0.05. Duplicate and/or triplicate observations were employed for the statistical evaluation. The statistical analysis was performed with IBM SPSS Statistics 28.

A principal component analysis (PCA) was used to evaluate the similarities among seeds and protein isolates from different harvest years and locations and the main drivers of variations among the samples regarding their protein content, ash content, protein recovery, flavor components, fatty acid compositions, LOX activity, and functional properties using the software OriginPro 2022.

## 3. Results and Discussion

### 3.1. Impact of Harvest Year and Location on Mass Yield and Protein Recovery

The protein recovery results indicated that the 2018 harvest had the lowest recovery, with no significant variations (*p* > 0.05) among the samples from other years. This lower recovery in 2018 could be attributed to aging effects and gradual protein degradation during storage, which might have impacted the extractability and functionality of the proteins, despite careful storage conditions. It could also be due to weather conditions, with 2018 being exceptionally warm and low in rain. Location-based differences were also minor, with the highest recovery observed from Location 1. This suggests that while the location may influence protein recovery, the effect is relatively modest. The slight differences in recovery across locations may be due to variations in soil nutrients, which can impact the overall protein content and composition in peas.

Regarding mass yield, significant discrepancies (*p* < 0.05) were found among the harvest years, with the highest yield recorded in 2020 and the lowest in 2022. These variations may be linked to differences in growing conditions, such as rainfall, temperature, and soil fertility, which affect the overall biomass production and, consequently, the mass yield. For example, the lower yield in 2022 might be due to environmental stresses that could have reduced overall crop productivity. Location-based differences were more pronounced, with peas from Location 2 yielding significantly higher mass (*p* < 0.05) than those from Location 1. This variation in mass yield is likely influenced by soil composition and fertility levels at different locations, where peas from Location 2 may have benefited from more favorable nutrient availability, enhancing overall biomass production. These variations in protein recovery and mass yield are likely related to differences in the crude composition of the peas, such as their protein and fiber content, as well as protein composition affected by harvest year and location (see Table 1 and Appendix A). The similarity in mass yield between Location 1 and the 2022 harvest suggests that both harvest year and location can influence protein recovery and mass yield. However, the impact on protein recovery appeared to be less pronounced, as location and year had minimal effects on the overall recovery percentages.

Overall, the findings indicate that cultivation location had a greater impact on protein yield than harvest year, likely due to the primary influence of soil composition (such as mineral and nutrient content) over weather conditions. This emphasizes the importance of considering both environmental factors and storage duration when analyzing protein recovery and mass yield.

### 3.2. Proximate Compositions and Surface Color

The variations in proximate composition of the pea flours and protein isolates from different harvest years and cultivation locations are reported in Appendix A. Both harvest year and location considerably affected the composition of the pea flours and their respective protein isolates. However, the efficiency of dehulling and grinding, with recorded rates of around 85% and 98%, respectively, did not exhibit significant (*p* > 0.05) differences across the samples. This finding implies that variations in harvest year and location do not substantially influence the effectiveness of dehulling and milling processes. On the other hand, harvest year and location impacted ash and moisture content in both pea flours and protein isolates. Ash content was notably higher in the protein isolates than in flours, ranging between approx. 1–3% in the pea flours and 4–7% in the protein isolates. This variation may be attributed to the selective recovery of some minerals such as zinc which can bind to proteins. The highest ash content was found in the proteins of pea samples harvested in 2020 and Location 1, which could be linked to their shared location.

In 2022, the harvested pea sample exhibited the highest moisture content, while the lowest content was observed in the oldest pea batch harvested in 2018. The sample from Location 1 displayed lower moisture than its counterpart from Location 2, indicating differing cultivation location effects on peas. These findings underscore the influence of weather conditions during cultivation, which were colder with higher precipitation and humidity in Location 1 compared to Location 2 (see Appendix A), as well as the influence of soil conditions (see Appendix A) on pea composition, as it was reported by Nikolopoulou et al. (2007) [10] that the environmental parameters, specifically the prevailing climate conditions and soil characteristics during the growth season can affect the pea seed composition. The lower ash content in the pea protein from Location 2, despite its soil composition, can be attributed to a combination of factors, including lower calcium availability, nutrient interactions, and the potential impact of soil pH on mineral uptake. These factors may have limited the overall mineral absorption by the plants, leading to a reduced ash content in the final protein isolate.

Flours displayed a relatively lower fat content compared to protein isolates. This increase in fat content in protein isolates can be attributed to the composition of the soil and its phosphorus content at the cultivation site as it appeared to influence the fat content of peas but also other key components such as non-starch polysaccharides, total tannins, and phytic acid levels [10]. Interestingly, harvest year and location significantly influenced the fat content of the flours, while the protein isolates remained relatively unaffected. This observation highlights the dynamic nature of environmental factors such as climate, soil composition, and agricultural practices in influencing the nutritional composition of pea products.

As shown in Table 1, the protein content in pea flours ranged from 18% to 23%, while the protein isolates exhibited significantly higher concentrations of 79% to 90%. This demonstrates the high efficiency of wet fractionation in extracting and concentrating proteins from peas. Harvest year showed a minor but significant (*p* < 0.05) effect on the protein content in the pea flours, as seen in the higher protein content in the flours from 2018 which was not visible in the protein isolate from 2018. Sweden’s climate experienced a very long drought season with high temperatures in 2018, affecting the composition of the pea seeds harvested that year, which represented an exceptional harvest year. Harvest location, however, showed a significant (*p* < 0.05) impact on the protein content in the pea flours and the protein isolates. This can likely be attributed to the environmental conditions and soil compositions experienced throughout the growth season and at the two locations [10]. Location 1 generally experienced colder temperatures throughout the year, with higher precipitation and humidity levels than Location 2, which had milder temperatures and less precipitation. Both locations faced higher wind speeds and fewer sunbeam hours during the colder months. Therefore, optimizing value chains to maximize and standardize protein quality from peas should not only focus on genetic factors but also consider environmental and geographical parameters. Climate conditions and soil characteristics during the growth season can lead to variations in the composition of pea seeds, the raw materials for protein extraction, and the final pea proteins.

Harvest year and location had significant (*p* < 0.05) effects on the color of the pea flours and protein isolates as presented in Appendix A. Moreover, the color might be influenced by the fat content, particularly after lipid oxidation by lipoxygenase [17]. The samples associated with the highest yellowness value were the pea flour and protein isolates harvested in 2018, while the lowest value of yellowness was found in the 2019 harvested samples. Meanwhile, the differences between the 2020 and 2022 harvested samples were minimal. In terms of examining the influence of location on the color of the samples, the pea flours sourced from Location 1 were associated with the highest yellowness value. Among the protein isolates, however, it was those from Location 2 that exhibited the greatest value of yellowness.

### 3.3. Impact of Harvest Year and Location on Lipoxygenase (LOX) Activity

According to Gao et al. (2020a) [28], the compounds responsible for the beany flavor are either inherent to peas or emerge from the degradation of fatty acids during storage and processing. LOX is naturally present in peas and becomes active immediately upon the grinding of the peas into flour. The peak activity of lipoxygenase occurs after the flour is dispersed in an aqueous medium, particularly when the pH is around neutral or slightly basic [14,29]. This dispersion process in an aqueous phase has the potential to activate specific enzymes such as LOX and lyase. These enzymes can then target unsaturated fatty acids, resulting in the production of lipid hydroperoxides and volatile compounds, including aldehydes and ketones [30]. Regardless of harvest year and location, LOX activity in the pea protein isolates was substantially lower than that of the pea flours (Figure 1). This could show that the classic alkaline extraction process followed by isoelectric precipitation has resulted in either the deactivation of lipoxygenase or its removal [31]. This is also in line with our previous observation on LOX activity in proteins isolated from six different varieties of field peas [32].

The 2022 harvest sample exhibited the highest LOX activity, while the oldest sample from 2018 showed the lowest activity. This finding underscores the impact of seed age on the reduction in LOX activity. Trindler et al. (2022) [4] reported that enzymes such as LOX and peroxidase lose their activity during storage. Therefore, it is difficult to extract the effect of harvest year per se on the LOX activity using the results from this study since the samples from different years have experienced different storage times before the analysis anyhow. Both pea flours and protein isolates from the two locations had lower LOX activity than the samples from different harvest years. Notably, protein isolates from Location 2 exhibited lower LOX activity than those from Location 1. Conversely, samples from Location 1 showed LOX activity more closely resembling that of the 2022 harvest, likely due to their shared origin from the same year and location.

### 3.4. Impact of Harvest Year and Location on Fatty Acid Composition

The compositions of the selected fatty acids, linoleic, linolenic, oleic, palmitic, etc., in the pea flours and protein isolates are presented in Table 2. The focus is mostly on two specific fatty acids, namely linoleic and linolenic acids, since the major products from lipid oxidation in legumes are those arising from the oxidation of these two fatty acids, as previously mentioned and documented by Trindler et al. (2022) [4].

Across all samples, the total amount of fatty acids increased in the protein isolates compared with their counterpart flours. This could be associated with the overall fat content, which was also higher in the protein isolates than the pea flours, as shown in Appendix A. As previously explained, this is likely due to the protein–lipid interactions that occur during the extraction process [17] and their coextraction and up-concentration with proteins during the removal of starch and fibers. Furthermore, the samples primarily consisted of unsaturated fatty acids, as reported by Trindler et al. (2022) [33]. The unsaturated fatty acids were predominantly polyunsaturated fatty acids (PUFAs), with levels ranging from 35.3% to 78.3% in pea flours and 60.9% to 79.5% in protein isolates. The samples also contained monounsaturated fatty acids (MUFAs), which spanned 9.0% to 32.3% in the pea flours and 4.7% to 9.6% in the protein isolates. Meanwhile, saturated fatty acids ranged from 12.7% to 44.9% in the pea flours and 13.2% to 30.3% in the protein isolates.

The linoleic acid content in pea flours from different harvest years and locations ranged from 28.4% to 56.6%, while protein isolates contained between 51.6% and 74.6%. The lowest linoleic acid content in pea flours was recorded in the 2018 harvest, whereas the 2020 harvest had the lowest content among protein isolates. Conversely, the highest linoleic acid content in protein isolates was found in the 2022 harvest, likely due to fatty acid degradation during storage, as content decreased with sample age, paralleling LOX activity trends. Additionally, protein isolates from Location 1 exhibited higher linoleic acid content than those from Location 2. Linolenic acid was detected in both pea flours and protein isolates, with contents ranging from 2.2% to 6.8% in flours and 3.7% to 6.8% in the isolates. The highest linolenic acid content was found in the protein isolates from the 2019 harvest and Location 2, while the lowest was observed in the 2022 harvest and Location 1. Oleic acid concentrations varied widely in pea flours (0.9% to 29.65%) but were narrower in the protein isolates (0.49% to 1.2%). The highest oleic acid content in the protein isolates was recorded for the 2020 harvest and Location 2, while the 2022 harvest and Location 1 had the highest content in flours. Comparatively, the protein isolates contained higher levels of palmitic acid than pea flours, ranging from 7.5% to 12.3% for isolates and 2.5% to 8.7% for flours. The highest palmitic acid content in the harvest year samples was noted in the protein isolates from 2019, while the lowest was in those from 2022. Furthermore, protein isolates from Location 2 showed higher palmitic acid content than those from Location 1. The higher palmitic acid content in the protein isolates from Location 2 compared to Location 1 can be attributed to several interconnected factors. Firstly, soil nutrient levels played a significant role; Location 2 exhibited higher potassium content (15% vs. 10%) and a lower pH (6.5 vs. 8), both of which can enhance lipid biosynthesis, including the synthesis of saturated fatty acids like palmitic acid. Secondly, variations in environmental conditions such as temperature, light exposure, and water availability may have influenced lipid metabolism, as plants under stress often produce more saturated fatty acids to stabilize their cellular membranes. Lastly, metabolic variations specific to the genetic and biochemical responses of the pea plants to their growing environment could have triggered alterations in the enzymes responsible for fatty acid synthesis, leading to the observed increase in palmitic acid content in Location 2. Thus, the combination of these factors likely contributed to the differences in fatty acid profiles between the two locations.

Variations in fatty acid compositions among pea seeds from different harvest years and locations can be attributed to differing weather conditions. Additionally, the formation of fatty acid breakdown products may increase with seed age, either enzymatically or through autoxidation, as detailed in previous sections.

### 3.5. Volatile Off-Flavor Compounds

The content of 11 aroma compounds previously identified as indicators of beany flavor in pea flours and protein isolates from various harvest years and locations is summarized in Table 3. Notably, volatile compound concentrations were found to be remarkably up-concentrated by a factor of over 10^3^–10^4^ in protein isolates compared to pea flours, with hexanal, 1-hexanol, 1-pentanol, 1-nonanol, and 3-methyl-1-butanol exhibiting the highest quantities.

Hexanal, a significant compound in protein isolates, is linked to the enzymatic or autoxidative degradation of unsaturated fatty acids and is the most prevalent aldehyde in peas [34]. The concentration of hexanal in pea flours followed the order 2018 > 2019 > 2020 > 2022, indicating that seed age significantly impacts volatile formation. Among protein isolates, those harvested in 2022 displayed the highest hexanal concentration and overall volatile content. This increase correlated with LOX activity (Figure 1) and fatty acid composition, particularly linoleic acid (Table 2). In contrast, the protein isolate from the 2018 harvest had the lowest hexanal content. Additionally, the protein isolate from Location 1 contained more hexanal than that from Location 2, highlighting the role of lipid oxidation in forming these volatile compounds during extraction due to LOX activity.

Total volatile content significantly differed between flours and protein isolates from the two locations, indicating a substantial impact of geographic variability on beany flavors in pea proteins. Specifically, protein isolates from Location 2 exhibited lower hexanal levels and overall volatile content than those from Location 1, aligning with the lower LOX activity observed in Location 2.

Odor-active molecules such as hexanol, 1-pentanol, 1-octen-3-ol, (E,E)-2,4-nonadienal, and (E,E)-2,4-decadienal (Table 3) result primarily from the autooxidation of linoleic and linolenic acids [4]. These compounds were more concentrated in protein isolates than in flours. There appears to be a correlation between the content of specific volatiles, like hexanol, and LOX activity (Figure 1, Table 3). Notably, the relatively high presence of 1-nonanol, which emits a rose–orange aroma and is associated with the oxidation of linoleic acid [35], was consistently abundant across all samples, even surpassing hexanal. This observation departs from recent studies [4,15] and suggests that factors such as physical damage or post-harvest processing of peas might exert a potential influence on alcohols such as 1-nonanol. Moreover, this raises the possibility of interactions among short-chain organic compounds—including aldehydes, ketones, and alcohols—that could contribute to the development of off-flavors in pea protein (Zhang et al., 2020) [20], warranting further investigation.

In summary, the findings highlight the influence of diverse factors such as differing harvest years, aging, and geographical locations on the levels of specific beany volatile compounds in pea flour and their protein isolates. These outcomes hold importance in establishing the most suitable storage environments, taking into account various crop years, and factoring in the environmental and soil conditions during cultivation [10].

### 3.6. Non-Volatile Off-Flavor Compounds

A range of antinutritional components, including phenolic compounds, tannins, saponins, and phytate, are acknowledged for their role in contributing to undesirable taste perceptions in proteins derived from plants and legumes [27].

Saponins, as a noteworthy antinutrient, hold significance due to their role in contributing to the bitterness of peas [27]. Predominantly localized in the cotyledon, saponins are closely linked with the protein bodies present in legumes [26]. The total saponin content in both the pea flours and protein isolates is shown in Appendix A. It is observed that saponin exhibits up-concentration in the protein isolates compared with the flours, which is in line with the results from Subasi et al. (2024) [5] who also observed that saponins tend to concentrate in protein isolates due to their affinity for proteins, facilitated by hydrophobic and electrostatic interactions during extraction [27]. Saponins are predominantly found in the cotyledons and are known to interact with protein bodies in legumes, which explains their persistence and concentration in isolated protein fractions [26]. In terms of environmental factors, our findings indicate that neither harvest year nor location significantly impacted the saponin content in the pea protein isolates. This stability in saponin content, regardless of growth conditions, has been reported in other studies as well. For instance, it has been noted that saponin levels in legumes were largely genetically determined, with minimal fluctuation across different environmental conditions [36]. This suggests that breeding strategies may offer a more effective approach for mitigating saponin-related bitterness than altering agronomic practices.

Polyphenols and tannins, inherent compounds in plants, possess astringent and bitter characteristics and can interact with proteins, leading to their precipitation [26,27]. The changes in phenolic compounds were not significant (*p* > 0.05) in most of the protein isolates compared with their pea flour counterparts. The harvest year and location also did not influence the phenolic content of pea flours and protein isolates. Finally, the total tannin content, shown in Appendix A, reveals a consistent pattern of tannin reduction in the protein isolates across all samples compared with the flours. Specifically, the trend displays the highest tannin values within the protein isolates of the harvest 2020, while the pea flours and protein isolates of the harvest 2018 exhibited the lowest tannin content among the yearly samples, with a 122.2% magnitude of difference. Additionally, the tannic acid contents in the flours and protein isolates from Location 2 was significantly higher than those from Location 1 with a 17.1% magnitude difference. However, the effect of the harvest location on the tannin in the protein isolates was almost negligible. It can be concluded that the harvest year significantly influenced the tannin content of both flour and protein isolates. Although the harvest year and location did not significantly influence tannin levels in our pea protein isolates, other studies have reported some variability in tannin content based on environmental conditions. For example, Oluwatosin (1999) [37] demonstrated that polyphenol and tannin contents in legumes could vary depending on the growth region and climatic factors. However, in our study, the stability of tannin levels across different conditions suggests that intrinsic genetic factors might play a larger role in determining their presence in pea proteins.

### 3.7. Molecular Weight Distribution

The SDS-PAGE profile of the protein isolates from various harvest years and locations is depicted in Figure 2, revealing bands typically spanning a range of 10–100 kDa across all samples. Notably, bands below 75 kDa can be attributed to convicilin, previously denoted as the α-subunit of vicilin by other researchers [15]. These bands are located at approximately 47–50, 30–34, and 15 kDa, which is attributed to the high similarity between convicilin and vicilin. The hexameric protein legumin, boasting a molecular mass of 300–400 kDa, is composed of acid (Leg α) and basic (Leg β) polypeptides covalently linked by a disulfide bond. The SDS-PAGE profile indicated legumin’s subunits at around 37–40 kDa and 19–22 kDa, respectively. In addition, legumin and vicilin are generally considered the predominant protein components in peas [15]. Additionally, lower molecular weight bands (less than 15 kDa) likely signify albumin polypeptides, which were notably minimal or absent in the different protein isolates. A distinct band at around 94 kDa, acknowledged as pea seed lipoxygenase (LOX), was consistently present in all extracted protein samples. This contrasts with the findings of [15], where the LOX band was described as a light band. Here, the LOX band was distinctly visible and detectable in all samples from different harvest years and locations, albeit with variations among them. Also, the subunit at 10 kDa associated with albumin in pea proteins showed higher intensity in the samples from 2018 and 2019. This could also reflect a higher recovery of albumin in older samples.

In general, the electrophoretic profiles of the protein derived from the various harvest years and locations were similar in terms of the type of protein bands, but the intensity of protein bands related to convicilin, vicilin, and especially legumin, which could indicate their quantity and ratio, was different among the samples. The sample from 2020 showed a lower intensity of bands related to convicilin (≅70 KDa) and legumin (≅40 KDa) compared to the samples from other harvest years. Generally, among samples from the two locations, the sample from Location 1 showed a higher intensity in all bands.

### 3.8. Functional Properties

Emulsification is a key functional property of pea protein that influences its applicability in the food industry [35]. Appendix A presents the emulsion activity index (EAI) and emulsion stability (ES) values for various pea proteins. The highest EAI was observed in protein isolates from the 2018 harvest, while the lowest was recorded in those from the 2022 harvest. Among location samples, protein isolates from Location 2 showed a higher EAI than those from Location 1, with an 11.8% difference, similar to the 2.6% difference observed in the 2022 harvest isolates from the same location. Significant variations (*p* < 0.05) in EAI were noted among different harvest years, but not among the locations. Regarding emulsion stability, protein isolates from the 2022 harvest exhibited the highest emulsion stability, while those from 2018 showed the lowest. No significant differences (*p* > 0.05) were found in emulsion stability among location samples. Overall, neither harvest year nor location had a significant impact on the emulsion stability of the pea proteins.

The alterations in the rheological characteristics of the protein isolates, illustrated through the storage modulus (G’), are depicted in Figure 3, across different temperatures and time intervals. The storage modulus (G’) represents a material’s capacity to store energy and its solid behavior. Overall, the gelation process was initiated with an initial decrease in G’ for all protein isolate samples, indicating protein denaturation and its softening due to the breakage of hydrogen interactions by heat [26]. After this initial decline, the protein isolates exhibited a rise in G’ between 5 and 11 min (i.e., at temperatures ranging from 35 °C to 55 °C), signifying the formation of protein structures. When the temperature was elevated to 90 °C and maintained for 30 min, variations in G’ values were evident for all the protein isolates. Notably, among the harvest year samples, the protein isolates of the peas from 2020 displayed the highest G’ values during this stage. Similarly, among the location samples, both the protein isolates from Locations 1 and 2 showcased G’ values similar to the sample from 2020 but higher than the protein from the other harvest years, indicating their superior structure formation via disulfide bonds and gel formation capacity upon heating compared to the rest of harvest year samples. This trend was also upheld when the temperature was lowered from 90 °C to 20 °C during the cooling step at the end of the gelation process. The protein isolates from the 2020 harvest at Locations 1 and 2 retained the highest G’ values during this step. In contrast, the remaining protein isolates exhibited noteworthy fluctuations in their final G’ values at the end of the cooling step, with the protein isolates of the harvest year 2019 registering the lowest G’ value. Significant (*p* < 0.05) variations were present in the rheological properties among samples from diverse harvest years and different geographical locations. We hypothesize that this could be related to not only the difference in the protein purity of the samples (especially for the protein isolates from Locations 1 and 2) but also the differences in the ratio of different proteins in each sample as shown in our SDS-PAGE results (Figure 3), especially for the sample from 2020 which had the lowest content of convicilin.

### 3.9. Principal Component Analysis (PCA)

To better understand not only the variations between the samples from the different harvest years and locations but also the drivers of variations in the flavor and functionality of the seeds (flours) and their protein isolates, PCA was used. As can be seen in Figure 4A, the first two principal components could explain around 75% of the total variance (variations between the seed samples). The older seeds from 2018 and 2019 showed one cluster on the positive side of PC1 with a high association with beany flavor compounds. Seeds from 2020, 2022, and Location 1 also formed one cluster on the negative side of PC1 with low levels of flavor components but high amounts of unsaturated fatty acids, especially linolenic and linoleic acids, and LOX activity. These results also support the hypothesis that the variations in the composition, LOX activity, and flavor of the seeds from the different harvest years are mostly driven by their age and degradation of fatty acids and their oxidation over time resulting in the formation of volatile beany compounds. The seed from Location 2 also showed a completely different cluster, which again shows the big impact of cultivation location on the properties of pea seeds.

The PCA results for the protein isolates (Figure 4B) where the two principal components could explain 65% of the variance showed that the protein isolated from years 2018 and 2019 formed one cluster, the samples from 2022 and Location 1 formed another cluster, and the sample from 2020 was in between the two clusters. Samples from 2022 and Location 1 were from similar years and locations and ended up very close to each other on the PCA graph. Again, the protein isolates from Location 2 showed a completely different cluster compared to all the other samples. Protein isolates from 2022 and Location 1 showed a high association with high levels of LOX activity and some volatile beany flavor compounds such as hexanal, 2,4-nonadienal, and 1-nonanol but a reverse association with linolenic acid content. This could also show the higher rate of formation of these volatile compounds due to the activity of LOX or autooxidation during the protein extraction using the pH-shift method which has been widely reported before, especially for the correlation between LOX activity and hexanal formation in pea proteins [4].

## 4. Conclusions

Both harvest year and location significantly (*p* < 0.05) affected protein recovery, with a minimal impact on mass yield. Harvest location influenced protein purity, while both year and location had a small but significant effect on crude composition. Differences in fatty acid composition, influenced by storage, affected lipid oxidation and off-flavor formation. The order of hexanal concentration in pea flours across different harvest years was 2018 > 2019 > 2020 > 2022, suggesting the significant impact of seed age on the formation of volatiles. Furthermore, the content of volatile beany compounds in the protein isolates from different harvest years was driven by their LOX activity. In addition, the total volatile content was notably different between the protein isolates from the two different locations, correlating with their PUFA content and LOX activity, which indicated the substantial influence of cultivation location on the development of beany flavors in pea proteins. The saponin and phenolic contents in both pea flours and protein isolate did not exhibit significant variations based on the harvesting years and cultivation locations. However, tannin content in the pea flours was notably affected by different harvesting years, but this effect was less pronounced in the protein isolates, which highlights the importance of considering cultivation conditions in mitigating undesirable taste perceptions. Harvest year and location did not affect emulsification capacity but did influence rheological properties, likely due to differences in protein purity and composition. These findings demonstrate the substantial impact of cultivation conditions on pea protein isolates and underscore the need for the industry to adapt extraction processes and formulations to manage variability.

## Figures and Tables

**Figure 1 foods-13-03423-f001:**
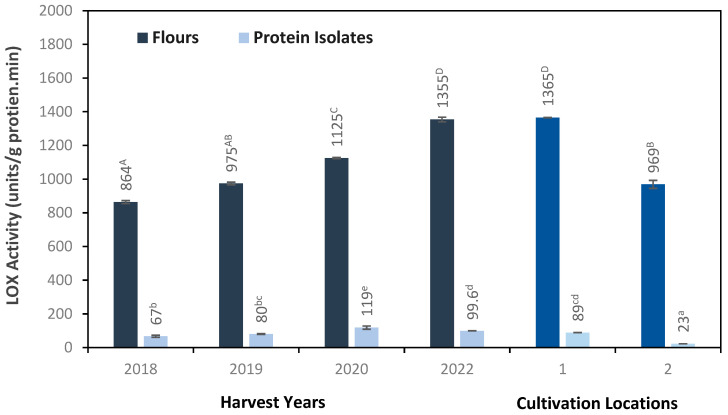
Lipoxygenase activity (units × (g/min)) of pea flours and protein isolates from different harvesting years and cultivation locations. Values are reported as mean ± SD, (n = 2). Ddifferent small and uppercase letters show significant different (*p* < 0.05) for protein isolate and pea flours, respectively.

**Figure 2 foods-13-03423-f002:**
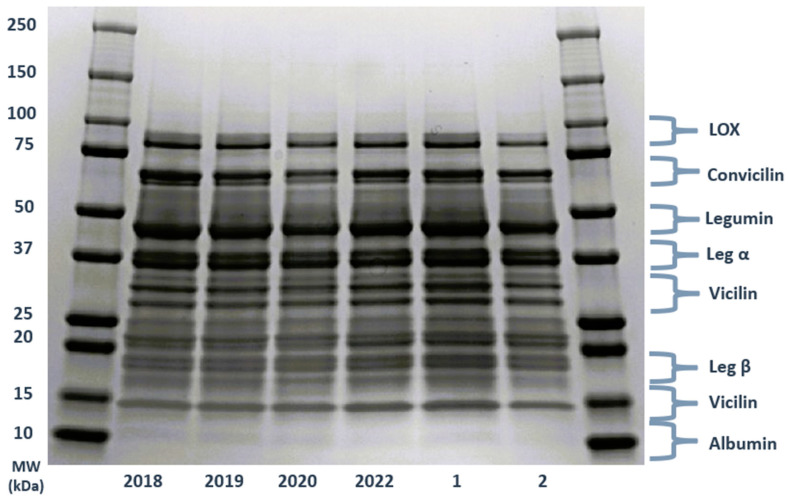
Molecular weight distribution of pea protein isolates from different harvesting years and cultivation locations as determined by SDS-PAGE. The gel imaging was carried out using Bio-Rad’s Gel Doc 2000. Abbreviations: Leg α: legumin α; Leg β: legumin β; and LOX: lipoxygenase. MW: molecular weight standard indicated in kilodalton (kDa).

**Figure 3 foods-13-03423-f003:**
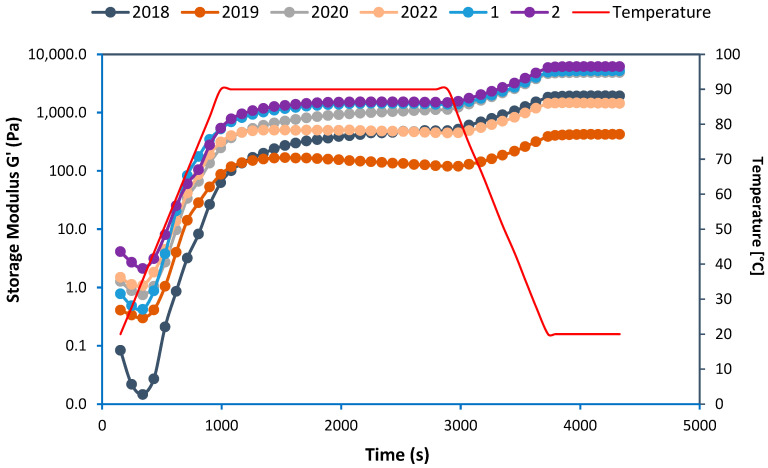
Rheological behaviors (i.e., storage modulus G’) of protein isolates from different harvest years and cultivation locations during in situ gelation via temperature ramp test including an initial heating step (5 °C/min, from 20 to 90 °C), followed by an isothermal step (90 °C, 30 min) and a final cooling step (5 °C/min, from 90 to 20 °C).

**Figure 4 foods-13-03423-f004:**
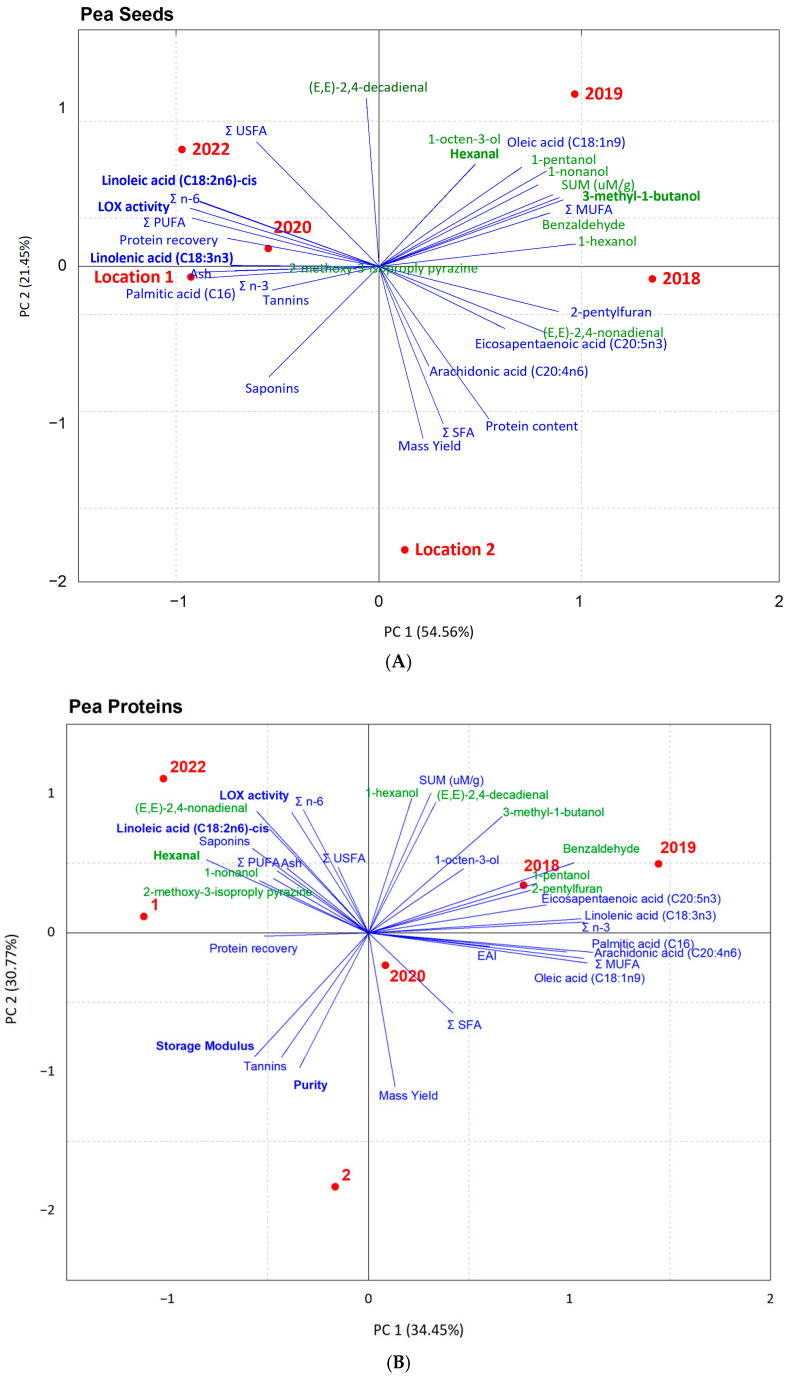
Principal component analysis biplots of the pea seeds/flours (**A**) and the protein isolates (**B**) from different harvest years (2018, 2019, 2020, and 2022) and cultivation locations (1 and 2). Green color shows volatile flavor compounds, red color shows harvest years and location and blue shows the rest of parameters.

**Table 1 foods-13-03423-t001:** Impact of harvest year and cultivation location on protein recovery, mass yield, and protein content from pea seeds from similar varieties.

	Protein Recovery [%]	Mass Yield [%]	Protein Content (%)	
			Seed	Protein Isolate
Harvest years				
2018	57.1 ± 1.3 ^a^	14.9 ± 0.3 ^ab^	22.8 ± 0.4 ^a^	81 ± 0.5 ^b^
2019	61.7 ± 0.7 ^b^	14.7 ± 0.1 ^ab^	19.0 ± 0.2 ^a^	80 ± 1.3 ^a^
2020	65.2 ± 0.9 ^b^	15.0 ± 0.2 ^c^	18.4 ± 0.1 ^a^	80 ± 0.3 ^a^
2022	61.6 ± 1.0 ^b^	14.1 ± 0.3 ^a^	18.1 ± 0.4 ^a^	79 ± 0.8 ^a^
Cultivation locations				
1	65.1 ± 0.8 ^b^	14.9 ± 0.2 ^a^	19.3 ± 0.1 ^a^	84.3 ± 0.3 ^a^
2	60.7 ± 1.4 ^a^	16.3 ± 0.4 ^b^	23.4 ± 0.4 ^b^	89.1 ± 0.2 ^ab^

Results are expressed as means ± standard deviation (n = 2). ^a–c^ Values with different small letters in each column are significantly different (*p* < 0.05).

**Table 2 foods-13-03423-t002:** Selected fatty acid compositions of pea flours and protein isolates from different harvesting years and cultivation locations.

	Harvest Year								Cultivation Location			
	2018		2019		2020		2022		1		2	
	Flour	Protein	Flour	Protein	Flour	Protein	Flour	Protein	Flour	Protein	Flour	Protein
Linoleic acid (C18:2n6)-cis	28.32 ± 0.0 ^d^	55.93 ± 1.51 ^ab^	36.63 ± 0.0 ^c^	65.07 ± 4.15 ^a^	47.64 ± 0.6	51.56 ± 5.26 ^b^	26.27 ± 1.97 ^d^	26.27 ± 1.97 ^d^	56.61 ± 1.07 ^A^	22.12 ± 0.0 ^C^	35.98 ± 5.0 ^C^	50.21 ± 1.72 ^B^
Linoleic acid (C18:2n6)-trans	0.05 ± 0.0 ^c^	--	0.27 ± 0.06 ^b^	0.25 ± 0.0 ^b^	--	--	44.03 ± 0.0 ^a^	44.03 ± 0.0 ^a^	--	52.52 ± 3.96 ^A^	0.47 ± 0.0 ^B^	0.18 ± 0.0 ^C^
Linolenic acid (C18:3n3)	2.20 ± 1.82 ^b^	5.43 ± 0.72 ^ab^	3.52 ± 1.47 ^b^	6.76 ± 0.26 ^a^	6.24 ± 0.0 ^a^	5.91 ± 0.06 ^a^	3.71 ± 0.22 ^b^	3.71 ± 0.22 ^b^	6.81 ± 0.26 ^A^	3.82 ± 0.10 ^B^	3.68 ± 0.7 ^B^	4.28 ± 0.12 ^B^
Arachidonic acid (C20:4n6)	0.05 ± 0.01 ^ab^	0.03 ± 0.0 ^c^	0.06 ± 0.01 ^a^	0.04 ± 0.0 ^b^	0.05 ± 0.0	0.02 ± 0.0 ^c^	0.02 ± 0.0 ^c^	0.02 ± 0.0 ^c^	0.07 ± 0.0	0.02 ± 0.00 ^B^	0.07 ± 0.0 ^A^	0.03 ± 0.0 ^B^
Eicosapentaenoic acid (C20:5n3)	0.16 ± 0.05 ^a^	0.08 ± 0.0 ^c^	0.16 ± 0.03 ^a^	--	0.12 ± 0.0 ^b^	--	0.03 ± 0.0 ^d^	0.03 ± 0.0 ^d^	0.15 ± 0.0 ^A^	0.04 ± 0.0 ^C^	0.15 ± 0.0 ^A^	0.07 ± 0.0 ^B^
Palmitic acid (C16)	3.67 ± 3.43 ^d^	10.57 ± 0.79 ^b^	5.32 ± 5.08 ^d^	12.25 ± 0.22 ^a^	8.66 ± 2.6 ^bc^	10.62 ± 0.09 ^b^	7.46 ± 0.85 ^c^	7.46 ± 0.85 ^c^	7.01 ± 0.07 ^B^	7.60 ± 0.64 ^AB^	6.72 ± 0.1 ^B^	9.58 ± 1.49 ^A^
Oleic acid (C18:1n9)	10.42 ± 8.16 ^b^	0.87 ± 0.21 ^d^	29.65 ± 0.0 ^a^	1.22 ± 0.05 ^c^	0.95 ± 0.1 ^d^	0.74 ± 0.15 ^d^	0.49 ± 0.08 ^d^	0.49 ± 0.08 ^d^	0.99 ± 0.0 ^A^	0.52 ± 0.10 ^B^	0.94 ± 0.2 ^A^	0.87 ± 0.04 ^A^
Σ SFAs	29.77 ± 13.95 ^a^	26.75 ± 3.07 ^a^	16.99 ± 9.12 ^b^	16.87 ± 0.39 ^b^	29.84 ± 2.7 ^a^	30.29 ± 8.62 ^a^	15.38 ± 4.15 ^b^	15.38 ± 4.15 ^b^	12.66 ± 0.42 ^C^	13.21 ± 2.4 ^C^	44.93 ± 5.9 ^A^	25.73 ± 0.62 ^B^
Σ USFAs	59.88 ± 0.37 ^c^	73.25 ± 3.86 ^b^	78.68 ± 4.24 ^ab^	83.13 ± 5.2 ^a^	70.16 ± 1.8 ^b^	69.65 ± 7.76 ^ab^	84.62 ± 6.02 ^a^	84.62 ± 6.02 ^a^	87.34 ± 9.42 ^A^	78.16 ± 6.24 ^AB^	58.82 ± 7.4 ^B^	77.05 ± 1.30 ^A^
Σ MUFAs	24.63 ± 12.51 ^a^	9.56 ± 1.35 ^b^	29.62 ± 2.74 ^a^	9.36 ± 0.15 ^b^	10.66 ± 0.1 ^b^	8.72 ± 0.81	5.17 ± 0.1 ^c^	5.17 ± 0.1 ^c^	9.03 ± 1.49 ^A^	4.72 ± 0.05 ^C^	9.43 ± 1.3 ^A^	7.58 ± 0.57 ^B^
Σ PUFAs	35.25 ± 12.88 ^d^	63.69 ± 2.51 ^c^	49.06 ± 6.99 ^d^	73.76 ± 5.0 ^b^	59.50 ± 1.9 ^c^	60.93 ± 6.95 ^c^	79.45 ± 5.66 ^a^	79.45 ± 5.66 ^a^	78.31 ± 7.93 ^A^	73.45 ± 6.29 ^A^	49.39 ± 6.1 ^B^	69.47 ± 0.77 ^A^
Σ n-3	2.41 ± 1.82 ^b^	5.65 ± 0.73 ^a^	3.70 ± 1.49 ^b^	6.92 ± 0.26 ^a^	6.45 ± 0.0 ^a^	6.02 ± 0.06 ^a^	3.77 ± 0.19 ^b^	3.77 ± 0.19 ^b^	7.09 ± 0.25 ^A^	3.93 ± 0.1 ^C^	3.94 ± 0.7 ^BC^	4.46 ± 0.11 ^B^
Σ n-6	28.42 ± 0.01 ^d^	55.96 ± 1.51 ^b^	36.95 ± 0.07 ^c^	65.36 ± 4.28 ^a^	47.69 ± 0.6 ^b^	51.58 ± 5.26 ^b^	70.32 ± 1.97 ^a^	70.32 ± 1.97 ^a^	56.68 ± 1.07 ^B^	66.04 ± 4.66 ^A^	36.45 ± 4.8 ^D^	50.42 ± 1.81 ^C^

Values are means ± standard deviation (n = 2) and are reported as relative %. Different letters (^a–d^ and ^A–D^) indicate significant differences (*p* < 0.05) between harvest years and cultivation locations, respectively. SFAs, saturated fatty acids; UFAs, unsaturated fatty acids; MUFAs, monounsaturated fatty acids; PUFAs, polyunsaturated fatty acids; n-3 = omega-3 fatty acids; n-6 = omega-6 fatty acids.

**Table 3 foods-13-03423-t003:** Selected off-flavor volatile compounds of pea flour and protein isolate from the 4 different harvest years and 2 locations.

	Harvest Year								Cultivation Location			
	2018		2019		2020		2022		1		2	
	Flour	Protein	Flour	Protein	Flour	Protein	Flour	Protein	Flour	Protein	Flour	Protein
Hexanal	0.216 ± 0.001	5.487 ± 0.006	0.204 ± 0.001	5.580 ± 0.002	0.142 ± 0.006	7.094 ± 0.01	0.103 ± 0.002	7.320 ± 0.002	0.104 ± 0.006	7.151 ± 0.01	0.144 ± 0.001	5.442 ± 0.003
1-hexanol	0.202 ± 0.002	8.127 ± 0.008	0.188 ± 0.001	7.297 ± 0.005	0.130 ± 0.002	7.3356 ± 0.004	0.097 ± 0.001	7.432 ± 0.001	0.094 ± 0.002	7.238 ± 0.004	0.135 ± 0.0	5.265 ± 0.004
1-octen-3-ol	0.001 ± 0.0	0.118 ± 0.0	0.002 ± 0.0	0.055 ± 0.0	0.001 ± 0.0	0.039 ± 0.0	0.001 ± 0.0	0.045 ± 0.0	0.001 ± 0.0	0.044 ± 0.0	0.001 ± 0.0	0.024 ± 0.0
Benzaldehyde	0.006 ± 0.0	0.318 ± 0.0	0.007 ± 0.0	0.422 ± 0.001	0.005 ± 0.0	0.2312 ± 0.0	0.003 ± 0.0	0.170 ± 0.0	0.002 ± 0.0	0.131 ± 0.0	0.004 ± 0.0	0.108 ± 0.0
2-pentylfuran	0.047 ± 0.0	0.700 ± 0.001	0.043 ± 0.0	0.547 ± 0.0	0.033 ± 0.001	0.601 ± 0.0	0.033 ± 0.0	0.679 ± 0.0	0.038 ± 0.0	0.780 ± 0.002	0.042 ± 0.0	0.346 ± 0.001
1-pentanol	0.200 ± 0.001	9.919 ± 0.01	0.219 ± 0.0	11.103 ± 0.0	0.153 ± 0.002	9.765 ± 0.002	0.120 ± 0.001	8.646 ± 0.001	0.108 ± 0.003	7.851 ± 0.003	0.113 ± 0.003	4.927 ± 0.003
1-nonanol	1.363 ± 0.01	59.815 ± 0.003	1.477 ± 0.04	51.043 ± 0.007	1.167 ± 0.04	47.540 ± 0.03	0.853 ± 0.005	63.499 ± 0.03	0.555 ± 0.02	34.010 ± 0.009	0.818 ± 0.02	23.956 ± 0.044
2-methoxy-3-isoproply pyrazine	0.002 ± 0.0	0.0442 ± 0.0	0.002 ± 0.0	0.0438 ± 0.0	0.002 ± 0.0	0.042 ± 0.0	0.002 ± 0.0	0.041 ± 0.0	0.002 ± 0.0	0.046 ± 0.0	0.002 ± 0.0	0.038 ± 0.0
(E,E)-2,4-nonadienal	0.045 ± 0.001	0.6641 ± 0.001	0.042 ± 0.0	0.545 ± 0.0	0.033 ± 0.001	0.596 ± 0.0	0.034 ± 0.0	0.6642 ± 0.0	0.038 ± 0.001	0.760 ± 0.002	0.043 ± 0.001	0.351 ± 0.001
(E,E)-2,4-decadienal	0.001 ± 0.0	0.052 ± 0.0	0.001 ± 0.0	0.0382 ± 0.0	0.001 ± 0.0	0.0382 ± 0.0	0.001 ± 0.0	0.042 ± 0.0	0.001 ± 0.0	0.030 ± 0.0	0.000 ± 0.0	0.011 ± 0.0
3-methyl-1-butanol	0.204 ± 0.0	9.824 ± 0.008	0.189 ± 0.003	10.149 ± 0.002	0.131 ± 0.003	9.003 ± 0.001	0.114 ± 0.004	7.985 ± 0.004	0.114 ± 0.004	7.392 ± 0.02	0.117 ± 0.001	4.641 ± 0.004
SUM (uM/g)	2.29 ± 0.01 ^C^	95.1 ± 0.6 ^c^	2.38 ± 0.04 ^C^	86.82 ± 0.2 ^b^	1.80 ± 0.04 ^B^	82.29 ± 0.2 ^a^	1.36 ± 0.003 ^A^	96.53 ± 0.5 ^c^	1.06 ± 0.02 ^A^	65.43 ± 0.99 ^b^	1.42 ± 0.02 ^A^	45.11 ± 1.2 ^a^

Results are represented in uM/g and as means ± standard deviation (n = 2). Different letters in the last row (^a–c^ and ^A–C^) indicate significant differences (*p* < 0.05).

## Data Availability

The data presented in this study are available on request from the corresponding author due to privacy and legal reasons.

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
