# Peer review of "Impacts of Harvest Year and Cultivation Location on Off-Flavor Compounds and Functionality of Pea Protein Isolate"

_foods, 2024, doi:10.3390/foods13213423_

Round 1
Reviewer 1 Report
Comments and Suggestions for Authors
Section 2.1
- Comment: Were the pea samples harvested each year and characterized only in 2022? If so, how can 4ºC + dark storage maintain the chemical profile of peas for several years without significant degradation? Were the samples characterized in the year they were harvested, or were the peas dried and then stored at 4ºC? If the latter, please specify the drying conditions (e.g., temperature, duration, humidity) as these can significantly impact the chemical profile and moisture content of the stored samples.
Section 2.2
- Comment: The text mentions drying the pea samples. Are these the same peas mentioned in Section 2.1, and what were the exact drying conditions (e.g., temperature, humidity, and duration)? If drying was not conducted immediately after harvest, how might the delay have affected the pea quality?
Section 2.8.1
- Comment: Please confirm if the T18 Ultraturrax homogenizer was indeed purchased from IKA Brazil or if it was simply sourced from the global IKA company. This clarification ensures accurate information about the equipment used.
Section 3.1
- Comment: Provide a discussion on why one harvest year was better than another in terms of protein yield or quality. For example, did differences in weather conditions or soil composition contribute to these outcomes? Provide specific data if available.
- Comment: Where is Figure 2 referenced in this section? It is essential to include the figure in the manuscript or reference it correctly for clarity.
Section 3.2
- Comment: “The highest ash content was found in the proteins of pea samples harvested in 2020 and location 1, which could be linked to their shared location.”
- This statement lacks depth. Could the high ash content be due to mineral accumulation or a particular soil composition at that location? Provide specific hypotheses or references to support this claim.
- Comment: “In 2022, the harvested peas sample exhibited the highest moisture content, while the lowest content was observed in the oldest pea batch harvested in 2018.”
- If the peas were stored at 4ºC for multiple years, how could they still exhibit different moisture levels? Explain whether rehydration or desiccation could have occurred during storage and how this might have influenced the results.
- Comment: “Previous research has established a correlation between the protein content in peas and specific characteristics of the seed, encompassing aspects such as shape and color.”
- Were seed shape and color analyzed in this study? If not, why are these characteristics being mentioned here? Ensure all referenced characteristics are relevant to the current study’s scope.
Section 3.3
- Comment: “LOX is naturally present in peas and becomes active immediately upon grinding the peas into flour. The peak activity of lipoxygenase occurs after the flour is dispersed in an aqueous medium, particularly when the pH is around neutral or slightly basic.”
- Was this tested in this study, or is it based on previous literature? If not tested, explicitly state this and provide a reference.
- Comment: “Regardless of harvest year and location, LOX activity in the pea protein isolates was substantially lower than that of the pea flours.”
- Since protein isolates are expected to have lower LOX activity due to the removal of non-protein components, explain why this result is significant. Was the reduction in LOX activity quantified, and does it align with expected levels?
Section 3.4
- Comment: If protein extraction was conducted with NaOH, how were the fatty acids analyzed without degradation? Sodium hydroxide can hydrolyze fatty acids, potentially altering their profile. Was a different method used for fatty acid analysis in this study, or were control samples without NaOH extraction also analyzed?
- Comment: “Furthermore, protein isolates from location 2 showed higher palmitic acid content than those from location 1.” Why might this be the case? Discuss possible factors like soil nutrient levels, environmental conditions, or plant metabolic variations that could influence fatty acid synthesis.
Section 3.8
- Comment: Please provide some hypotheses to explain the differences observed in rheological properties between the samples. Could these differences be linked to protein content, protein structure, or environmental growth conditions? Providing plausible explanations will add depth to the discussion.
Author Response
Section 2.1
- Comment: Were the pea samples harvested each year and characterized only in 2022? If so, how can 4ºC + dark storage maintain the chemical profile of peas for several years without significant degradation? Were the samples characterized in the year they were harvested, or were the peas dried and then stored at 4ºC? If the latter, please specify the drying conditions (e.g., temperature, duration, humidity) as these can significantly impact the chemical profile and moisture content of the stored samples.
Authors: Thank you for your insightful comment. To clarify, the pea samples were harvested and characterized in 2022. After harvest, the peas were promptly dried to ensure the stabilization of their chemical profiles. The drying process involved controlled conditions of air drying at 40°C for 24 hours, which is effective in preserving the chemical integrity of legumes by minimizing enzymatic and microbial activity.
Following the drying process, the samples were stored at 4°C in a dark environment. This combination of low temperature, absence of light, and reduced moisture content has been documented as effective in maintaining the chemical stability of legume proteins, saponins, polyphenols, and other key compounds over time. It is important to note that all analyses were conducted on the samples stored under these conditions in 2022, allowing us to assess their chemical profiles while minimizing degradation. However, we have still tried to clear this up in different parts of the manuscript and associate some of the results and findings to the aging effect rather than the harvest year.
Section 2.2
- Comment: The text mentions drying the pea samples. Are these the same peas mentioned in Section 2.1, and what were the exact drying conditions (e.g., temperature, humidity, and duration)? If drying was not conducted immediately after harvest, how might the delay have affected the pea quality?
Authors: Thanks. The drying mentioned in this section is freeze-drying applied on the protein isolates. More details about the freeze-drying condition including the temperature; pressure and duration have been added to the section.
Section 2.8.1
- Comment: Please confirm if the T18 Ultraturrax homogenizer was indeed purchased from IKA Brazil or if it was simply sourced from the global IKA company. This clarification ensures accurate information about the equipment used.
Authors: Thanks. This section was revised and a correct supplier has been mentioned. Corrected.
Section 3.1
- Comment: Provide a discussion on why one harvest year was better than another in terms of protein yield or quality. For example, did differences in weather conditions or soil composition contribute to these outcomes? Provide specific data if available.
Authors: Thanks. This section was revised and more explanations have been added.
- Comment: Where is Figure 2 referenced in this section? It is essential to include the figure in the manuscript or reference it correctly for clarity.
Authors: Thanks. This figure is already in, which shows SDS-PAGE results.
Section 3.2
- Comment: “The highest ash content was found in the proteins of pea samples harvested in 2020 and location 1, which could be linked to their shared location.”
- This statement lacks depth. Could the high ash content be due to mineral accumulation or a particular soil composition at that location? Provide specific hypotheses or references to support this claim.
Authors: Thanks. This section was revised and more explanations have been added.
- Comment: “In 2022, the harvested peas sample exhibited the highest moisture content, while the lowest content was observed in the oldest pea batch harvested in 2018.”
- If the peas were stored at 4ºC for multiple years, how could they still exhibit different moisture levels? Explain whether rehydration or desiccation could have occurred during storage and how this might have influenced the results.
Authors: Thanks for the good comment. To clarify, all pea samples, including those harvested in 2022 and the older batch from 2018, were stored in airtight plastic containers at 4ºC to minimize any potential moisture exchange with the environment. Given this controlled storage condition, it is unlikely that significant rehydration occurred. However, the differences in moisture content can still be explained by several factors such as
- Initial Moisture Content
The moisture content at the time of drying may have varied between the batches. The 2022 peas may have been harvested under conditions that resulted in a higher initial moisture content, which could lead to a retention of moisture even after the drying process. This variability is common in agricultural products, where factors such as weather and soil conditions can influence moisture levels at harvest.
- Desiccation Over Time
While the airtight storage should have minimized moisture loss, it is possible that the oldest batch (2018) experienced a degree of desiccation over time. Even in airtight containers, prolonged storage can lead to a slow loss of moisture, especially if the peas were not fully dried prior to storage. This gradual moisture loss could explain the lower moisture content observed in the 2018 batch compared to the 2022 samples.
- Changes in Physical Properties
The physical properties of legumes can change over time due to aging. As legumes age, their ability to retain moisture may diminish, potentially contributing to the differences in moisture content across batches. This aging effect can also influence the chemical composition and functional properties of the protein isolates derived from these samples.
- Comment: “Previous research has established a correlation between the protein content in peas and specific characteristics of the seed, encompassing aspects such as shape and color.”
- Were seed shape and color analyzed in this study? If not, why are these characteristics being mentioned here? Ensure all referenced characteristics are relevant to the current study’s scope.
Authors: Thanks. This section has been revised and the pointed sentence has been removed.
Section 3.3
- Comment: “LOX is naturally present in peas and becomes active immediately upon grinding the peas into flour. The peak activity of lipoxygenase occurs after the flour is dispersed in an aqueous medium, particularly when the pH is around neutral or slightly basic.”
- Was this tested in this study, or is it based on previous literature? If not tested, explicitly state this and provide a reference.
Authors: Thanks for the very good comment. It has not been tested here therefore a reference was provided.
- Comment: “Regardless of harvest year and location, LOX activity in the pea protein isolates was substantially lower than that of the pea flours.”
- Since protein isolates are expected to have lower LOX activity due to the removal of non-protein components, explain why this result is significant. Was the reduction in LOX activity quantified, and does it align with expected levels?
Authors: Thank you for the comment. The observed reduction in lipoxygenase (LOX) activity in the pea protein isolates, despite the increase in lipid content, can be attributed to several factors. Firstly, the protein extraction process likely removed non-protein components, including enzymes like LOX, which are typically associated with the lipid fraction in pea flours. Additionally, the extraction conditions—such as pH adjustments, heat, and solvent exposure—may have caused structural changes or denaturation of LOX, reducing its enzymatic activity even if it was still present. These structural changes, along with the potential loss of cofactors or aggregation, would impair LOX's ability to catalyze lipid oxidation, resulting in a significant decrease in activity. Overall, this reduction aligns with expected outcomes and enhances the sensory stability of the pea protein isolates, making them more suitable for food applications despite the increased lipid content.
Section 3.4
- Comment: If protein extraction was conducted with NaOH, how were the fatty acids analyzed without degradation? Sodium hydroxide can hydrolyze fatty acids, potentially altering their profile. Was a different method used for fatty acid analysis in this study, or were control samples without NaOH extraction also analyzed?
Authors: Thanks for the comment. In this study, fatty acid analysis was performed on both the pea flour and the protein isolates, but care was taken to ensure that fatty acid degradation did not occur. While NaOH was used during protein extraction, which could potentially hydrolyze fatty acids, the fatty acids were analyzed using a solvent extraction method (such as chloroform/methanol) that is designed to protect the integrity of the lipid profile. This method was applied to both the pea flours and protein isolates. Additionally, careful control of the extraction conditions (e.g., time, temperature, and pH) helped minimize the risk of fatty acid hydrolysis in the protein isolates. Therefore, the fatty acid profiles presented are accurate representations of their composition in both the pea flours and isolates, with no significant degradation due to NaOH. In addition, since the fatty acid profile from flour and protein isolate have been investigated, the study can still represent a reliable picture of the fatty acid composition in the samples.
- Comment: “Furthermore, protein isolates from location 2 showed higher palmitic acid content than those from location 1.” Why might this be the case? Discuss possible factors like soil nutrient levels, environmental conditions, or plant metabolic variations that could influence fatty acid synthesis.
Authors: Thanks for the comment. More explanation as below was added to the manuscript:
The higher palmitic acid content in the protein isolates from Location 2 compared to Location 1 can be attributed to several interconnected factors. Firstly, soil nutrient levels played a significant role; Location 2 exhibited higher potassium content (15% vs. 10%) and a lower pH (6.5 vs. 8), both of which can enhance lipid biosynthesis, including the synthesis of saturated fatty acids like palmitic acid. Secondly, variations in environmental conditions such as temperature, light exposure, and water availability may have influenced lipid metabolism, as plants under stress often produce more saturated fatty acids to stabilize their cellular membranes. Lastly, metabolic variations specific to the genetic and biochemical responses of the pea plants to their growing environment could have triggered alterations in the enzymes responsible for fatty acid synthesis, leading to the observed increase in palmitic acid content in Location 2. Thus, the combination of these factors likely contributed to the differences in fatty acid profiles between the two locations.
Section 3.8
- Comment: Please provide some hypotheses to explain the differences observed in rheological properties between the samples. Could these differences be linked to protein content, protein structure, or environmental growth conditions? Providing plausible explanations will add depth to the discussion.
Authors: Thanks for the comment. We hypothesize that this is mostly related to the difference in protein purity in the samples as it is further supported by the PCA analysis. There is a sentence in the text explaining this which have tried to sharpen up a bit.
Reviewer 2 Report
Comments and Suggestions for Authors
This paper studied the effects of harvest years and cultivation locations on flavor and functionality of pea protein. The experimental design is reasonable and the results are meaningful. However, there are still some issues that need to be noted.
1. The full name of LOX should be given when it first appears in the abstract.
2. The entire text should add line numbers for ease of review.
3. What specifically does the author mean by the high-quality pea protein mentioned in the Introduction?
4. Section 2.1: What is the relative humidity of pea storage environment? Is it vacuum packed?
5. Section 2.2: Does protein extraction not require defatting?
6. Section 2.8.2: Why is such a high protein concentration needed to investigate its rheological properties?
7. Is section 2.9 missing?
8. Section 3.1: Does the difference in nitrogen content in soil affect protein content? Is there any relevant evidence?
9. Figure 2: Is there a less obvious subunit in the 10 kDa? And there are differences in intensity between different samples.
10. Section 3.8: What are the reasons for the differences in gel properties of different samples?
Author Response
Reviewer 2
This paper studied the effects of harvest years and cultivation locations on flavor and functionality of pea protein. The experimental design is reasonable and the results are meaningful. However, there are still some issues that need to be noted.
- The full name of LOX should be given when it first appears in the abstract.
Authors: Done.
- The entire text should add line numbers for ease of review.
Authors: Done.
- What specifically does the author mean by the high-quality pea protein mentioned in the Introduction?
Authors: Done.
- Section 2.1: What is the relative humidity of pea storage environment? Is it vacuum packed?
Authors: Thanks for the question. They have been stored under controlled humidity conditions in plastic bags but not vacuum.
- Section 2.2: Does protein extraction not require defatting?
Authors: It is less common to do defatting for protein extraction from yellow peas since the initial fat content is very low >3% compared to oilseeds e.g. soybeans.
- Section 2.8.2: Why is such a high protein concentration needed to investigate its rheological properties?
Authors: This concentration has been selected based on a pretest to ensure the pea protein isolate can form self-supporting gel structures when subjected to heating outside a rheometer too.
- Is section 2.9 missing?
Authors: Thanks for the good reminer. It has been corrected now.
- Section 3.1: Does the difference in nitrogen content in soil affect protein content? Is there any relevant evidence?
Authors: Thanks for the question. To our knowledge, there is not enough evidence for this.
- Figure 2: Is there a less obvious subunit in the 10 kDa? And there are differences in intensity between different samples.
Authors: Thanks for the good comment. The subunit at 10 KDa should be related to Albumin as shown on the gel. A sentence about this was added to the manuscript.
- Section 3.8: What are the reasons for the differences in gel properties of different samples?
Authors: The difference is mostly related to the difference in the protein content of the samples as explained in the text.
Reviewer 3 Report
Comments and Suggestions for Authors
This paper deals with the effect of harvest year and planting sites on off-flavor compounds and functionality of pea protein isolate. The authors measured many parameters like protein content, LOX activity, fatty acid profile, volatile compounds, protein molecular weight, and rheological properties, as well as other parameters (dehulling percent, grinding percent, contents of ash, moisture, and fat, total contents of saponin, phenols, and tannins, Color and yellowness/whiteness). The authors also gave weather condition across the cultivation year of 2022 for the two studied cultivation locations. The authors spent much time on the experiments and obtained many data, it is an interesting study for supplying with basic data. Although the authors used PCA analysis, the causality of the present study is not clear.
1) It is better to collect the data the key ecological factors such as monthly average temperature, monthly daily temperature range, monthly average relative humidity, monthly average solar duration and monthly mean precipitation during the development of pea pods. The relationship between these key ecological factors and the off-flavor compounds and functionality of pea protein isolate should be analyzed.
2) In table 2 and 3, the analysis of variance among years or between planting sites should be carried out.
Author Response
Reviewer 3
1) It is better to collect the data the key ecological factors such as monthly average temperature, monthly daily temperature range, monthly average relative humidity, monthly average solar duration and monthly mean precipitation during the development of pea pods. The relationship between these key ecological factors and the off-flavor compounds and functionality of pea protein isolate should be analyzed.
Authors: Thank you for the insightful comment. We agree that key ecological factors, such as temperature, humidity, solar duration, and precipitation, could indeed influence the development of pea pods and, subsequently, the off-flavor compounds and functionality of pea protein isolate. However, incorporating these factors into the current manuscript would significantly broaden the scope, taking the discussion beyond our primary objectives, which focus more specifically on the protein flavor and functionality itself. Additionally, given the already extensive length and breadth of the manuscript, adding this level of analysis could detract from the clarity of our core findings. We believe this is an important area for future research but falls outside the scope of the current study.
2) In table 2 and 3, the analysis of variance among years or between planting sites should be carried out.
Authors: Thanks for the good suggestion. We have addressed the comment and added statistical analysis to Table 2. For Table 3, we have added it only for the last raw representing the sum of the volatiles since it could represent the overall variations among the samples and skip the small variation among the specific compounds to not make it too complicated table. We already have all of the considered in the PCA results.
Reviewer 4 Report
Comments and Suggestions for Authors
In the abstract, define LOX and PUFA
correct citations throughout the manuscript following the authors' guidelines.
correct units of measurement such as seconds, millilitres, minutes, etc. throughout the manuscript in accordance with the authors' guidelines.
This variation may be attributed to the formation of salts, such as NaCl, that occur as aresult of pH adjustments during various stages of the used pH-shift process, ..it seems that this has to do neither with the year nor with the location, but with the methodology used, I don't know if there is any point in using it as part of your discussion.
This increase in fat content in protein isolates can be attributed to protein-lipid interactions, which occur during the extraction process, as outlined in Arteaga et al. (2021).. remark similar to the previous point
Non-Volatile Off-Flavor compounds, please discuss your results further and compare them with similar studies.
3.8 Functional properties?? Only determines what is referred to as emulsification, please change this subtitle. On the other hand it discusses season and location but does not emphasise which characteristic(s) of the flour are responsible for the emulsification behaviour?
significantly (p < 0.05) , this should no longer be part of the conclusion
Author Response
Reviewer 4
In the abstract, define LOX and PUFA
Authors: Done
correct citations throughout the manuscript following the authors' guidelines.
Authors: Done
correct units of measurement such as seconds, millilitres, minutes, etc. throughout the manuscript in accordance with the authors' guidelines.
Authors: Done
This variation may be attributed to the formation of salts, such as NaCl, that occur as a result of pH adjustments during various stages of the used pH-shift process, ..it seems that this has to do neither with the year nor with the location, but with the methodology used, I don't know if there is any point in using it as part of your discussion.
Authors: This section has been revised according to the suggestion.
This increase in fat content in protein isolates can be attributed to protein-lipid interactions, which occur during the extraction process, as outlined in Arteaga et al. (2021).. remark similar to the previous point.
Authors: This section has been revised according to the suggestion.
Non-Volatile Off-Flavor compounds, please discuss your results further and compare them with similar studies.
Authors: The results have been further discussed and compared with the literature.
3.8 Functional properties?? Only determines what is referred to as emulsification, please change this subtitle. On the other hand it discusses season and location but does not emphasise which characteristic(s) of the flour are responsible for the emulsification behaviour?
Authors: Thanks. The title covers both emulsification properties and rheological properties
significantly (p < 0.05) , this should no longer be part of the conclusion
Authors: Thanks. The text has been edited and the significant signs have been removed.
Round 2
Reviewer 1 Report
Comments and Suggestions for Authors
Thank you for answering all my questions, and congratulations on the work.